# Modeling of Protein Structural Flexibility and Large-Scale Dynamics: Coarse-Grained Simulations and Elastic Network Models

**DOI:** 10.3390/ijms19113496

**Published:** 2018-11-06

**Authors:** Sebastian Kmiecik, Maksim Kouza, Aleksandra E. Badaczewska-Dawid, Andrzej Kloczkowski, Andrzej Kolinski

**Affiliations:** 1Faculty of Chemistry, Biological and Chemical Research Center, University of Warsaw, Pasteura 1, 02-093 Warsaw, Poland; sekmi@chem.uw.edu.pl (S.K.); mkouza@chem.edu.uw.pl (M.K.); adawid@chem.uw.edu.pl (A.E.B-D.); 2Nationwide Children’s Hospital, Columbus, OH 43205, USA

**Keywords:** protein dynamics, coarse-grained simulation, Monte Carlo dynamics, structural flexibility, large-scale dynamics, elastic network model

## Abstract

Fluctuations of protein three-dimensional structures and large-scale conformational transitions are crucial for the biological function of proteins and their complexes. Experimental studies of such phenomena remain very challenging and therefore molecular modeling can be a good alternative or a valuable supporting tool for the investigation of large molecular systems and long-time events. In this minireview, we present two alternative approaches to the coarse-grained (CG) modeling of dynamic properties of protein systems. We discuss two CG representations of polypeptide chains used for Monte Carlo dynamics simulations of protein local dynamics and conformational transitions, and highly simplified structure-based elastic network models of protein flexibility. In contrast to classical all-atom molecular dynamics, the modeling strategies discussed here allow the quite accurate modeling of much larger systems and longer-time dynamic phenomena. We briefly describe the main features of these models and outline some of their applications, including modeling of near-native structure fluctuations, sampling of large regions of the protein conformational space, or possible support for the structure prediction of large proteins and their complexes.

## 1. Introduction

The biological activity of proteins involves adopting a specific conformation, their local fluctuations and, in many cases, structural transitions between different conformations. Structural flexibility can range from small side-chain fluctuations to large rearrangements of the entire protein backbone or its fragments, for example, disordered regions [1,2,3]. Most of the knowledge about protein structures comes from X-ray crystallography that, in most cases, offers only a static view of well-ordered protein regions. Determination of protein structure dynamics remains difficult using either experiment or computer simulations, despite significant effort in the field of drug design [4,5,6]. All-atom molecular dynamics (MD), the classical simulation technique, enables relatively inexpensive studies of small protein fluctuations (e.g., side-chain or very local backbone moves) and rather small protein systems [7]. In practice, many of protein systems are either too large to be effectively simulated using MD or require very large supercomputer resources. Some acceleration of MD can be achieved using so-called Go (or structure-based) models that provide quite valuable but limited insight into large-scale protein dynamics [8,9,10]. Significant computational speed-up is possible by simplifying the protein description to a less complex level than that of MD. This is offered by coarse-grained (CG) protein models [8,11] and elastic network models (ENM) used in conjunction with normal-mode analyses (NMA) [11,12,13,14]. Both approaches can be used as key components of multiscale modeling methods [15,16] merging computational tools of various resolutions from the low-resolution to atomistic level [17]).

In this review, we briefly discuss two representative CG simulation models of protein dynamics (Section 2.2 and Section 2.3) and various ENM-based modeling techniques (Section 2.4). Next, we review their applications to modeling protein local dynamics: structural flexibility of folded proteins (Section 3.1), and large-scale structural transitions (Section 3.2). The discussed approaches (Figure 1) differ in their levels of accuracy of protein structure representation, employed models of the interaction scheme, and their sampling of system dynamics. Interestingly, for many problems all these methods lead to surprisingly similar results, with one difference: the ENM approach is the simplest and therefore can be effectively used for studies of very large systems, obviously with all the limitations of structure-based models. Finally, we discuss the advantages and limitations of the presented approaches.

## 2. Coarse-Grained Protein Modeling

### 2.1. From All-Atom to Coarse-Grained Modeling

Molecular modeling of protein dynamics plays important roles in many branches of biophysics, molecular biology, and related sciences. The most popular strategies for modeling proteins and other biomolecular systems are classical MD simulation methods. MD, however, has several limitations [18]. Important biomacromolecular systems can be very large, and the timescale of interesting processes can be beyond the MD simulation range accessible to contemporary computers. Using the most powerful and dedicated computing devices, it is now possible to simulate structure assembly dynamics of small fast-folding proteins [19]. An alternative is to use CG modeling tools [8,20]. The CG modeling of proteins, initiated half a century ago by Levitt (and others) [15,16], has recently become a rapidly growing branch of molecular modeling. The general idea is to reduce the number of explicitly treated degrees of freedom in the modeled systems (Figure 2) and simplify energy calculations to speed up computations and thereby allow simulations of larger systems and/or longer processes.

Many CG approaches have been proposed, using different levels of reduced representations of biomacromolecules, different force-field schemes, and different strategies for sampling the conformational space [8,20,21,22,23]. We discuss the models that, in spite of their reduced numbers of explicitly treated atoms or pseudoatoms, allow realistic reconstruction of atomic details, and thereby open a possibility for the multiscale integrative modeling of protein systems. In addition, instead of the physics-based force field, we opted for statistical, knowledge-based models of interactions. Both approaches have their advantages and disadvantages, but since proteins (and generally biomacromolecules) can be characterized by many specific features, and owing to the rapidly growing databases of structural biology, statistical force fields can be a good choice. Consequently, instead of using various molecular dynamics schemes, we prefer properly designed Monte Carlo dynamics sampling. Such approaches prove to be very productive in the computational prediction of protein structures, modeling of protein dynamics, and molecular docking mechanisms [8].

Below, we discuss in more detail two representative models of various resolutions based on the above-outlined strategy of CG modeling: a relatively high-resolution Cα, Cβ, side-chain (CABS) [24] CG model (assuming C-Alpha, C-Beta, and side-chain representation of polypeptide chains) and a low-resolution Single United Residue per Preaveraged Secondary Structure fragment (SURPASS) [25] CG model. As illustrated in Figure 3, CG simplifications of the models can significantly increase (by orders of magnitude) both the size and timescale of tractable systems. Obviously, the deeper the coarse-graining of the modeled structures, the larger the systems that can effectively be simulated. On the other hand, coarse-graining of the studied structures always results in neglecting some subtle features. How costly it is to simplify the representation depends not only on the assumed level of coarse-graining, but also on the force-field model and sampling schemes used in simulations. CABS, SURPASS, or other related CG models can address a broad range of structure dynamics problems, including protein folding–unfolding or associations with other biomacromolecules, obviously with the above-mentioned computational limitations.

### 2.2. Coarse-Grained CABS Model

CABS [24] is a coarse-grained model for simulations of protein structures and dynamics (see Figure 4 and recent review on its design and applications [8]). CG representation of amino acid residues is reduced to up to four pseudoatoms or united atoms. These pseudoatoms of CABS models represent main-chain alpha carbons (CA), beta carbons (B), and the center of mass of side chains (S). The fourth pseudoatom is placed in the center of the Cα-Cα pseudobond and is used for the CG definition of the main-chain hydrogen bonds.

The force field of CABS is knowledge-based, derived from the statistical analysis of a representative set of known protein structures, although it also works surprisingly well for partially unfolded proteins [8,24]. The force field contains local conformational biases dependent on amino acid identities and excludes volumes of pseudoatoms, a model of strongly directional main-chain hydrogen bond attractions and contact potentials for side chains [26]. The latter have a specific feature, making the CABS force field very sensitive to the amino acid content in the studied proteins. Namely, this potential (for single-domain proteins) strongly depends on the mutual orientations of the contacting side chains [24,26]. For instance, when modeling a single-domain globular protein, oppositely charged side chains are strongly attractive if their contacts are parallel, and antiparallel contacts are repulsive. Indeed, when looking into globular proteins, oppositely charged residues usually come into contact on the protein surface, and consequently the corresponding side chains are almost parallel. On the contrary, contacts of charged residues in the central regions of a globule, where parallel and antiparallel orientations of side chains in contact are possible, are extremely rare, therefore the antiparallel contacts of charged residues are treated as repulsive (unlike). This orientation-dependent definition of side-chain contact interactions is a powerful way to introduce a crude account for implicit solvent effects in CG models.

The sampling scheme of the CABS model (constant temperature simulations, cooling/heating simulations, replica-exchange simulations, and other sampling strategies) is based on the MC dynamics method. The dynamics of the entire protein structure is modeled as a very long random sequence of small local moves, including single amino acid moves to small-fragment (composed of two, three, and rarely a slightly larger number of amino acids) local moves. MC dynamics provides very reasonable dynamics trajectories for long-time processes, although the time unit of such dynamics is not a priori defined. MC dynamics can be properly scaled by fitting (using mean-square displacements, or related parameters, as global motion criteria) fragments of the MC trajectories to corresponding MD trajectories and/or to appropriate experimental data. CABS MC dynamics simulations are very fast, not only due to the reduced representation of protein structures, but also thanks to the high-co-ordination lattice representation of the model structures. The underlying lattice enables use of precalculated data tables of local geometry and interactions. This makes computing new randomly selected structures and interactions extremely fast. On the other hand, the very high co-ordination number of interacting pseudoatoms (due to the high resolution of the underlying lattice) does not impose any noticeable directional biases that is so painful for lower-resolution lattice models of protein structures [27].

The CABS model has been successfully used in numerous studies, including the folding of globular proteins [28,29,30,31,32,33], dynamics of protein–peptide binding [34,35], flexibility of globular proteins [36,37,38,39,40], chaperonin mode of action [32], protein insertion into the biological membrane [41], structure prediction of proteins [42,43,44] and protein–peptide complexes [45,46,47,48,49]. The CABS has been implemented as a simulation component of several multiscale modeling tools, publicly available as web servers and standalone packages (see http://biocomp.chem.uw.edu.pl and http://lcbio.pl). The CABS-based tools are targeted onto specific simulation tasks, including simulations of near-native (or in the vicinity of another provided structure) protein flexibility [37,38,39,40], modeling of protein folding (including de novo or template-based structure prediction) [42,50], and unrestrained molecular docking of flexible peptides to flexible protein receptors [45,46,47,49]. Very recently, the CABS model has been made available as the CABS-flex standalone package [40], which is a flexible simulation environment for modeling protein dynamics including small-scale fluctuations (based on the input structure) or large-scale moves. The CABS-flex package offers easy modification of the CABS simulation parameters (and other modeling stages like scoring, structural clustering, or all-atom reconstruction) and using various input data (in the form of user-defined distance restraints). Therefore, the CABS-flex package can be easily incorporated into other multiscale modeling methodologies of structural biology. 

### 2.3. Coarse-Grained SURPASS Model

By implementing specific structural regularities characteristic to protein globules, it is possible to design deeply coarse-grained representations of protein chains that are much simpler than CABS (and other models of similar resolution), and which, regardless of their simplicity, maintain some of the most important features of real systems. SURPASS is an example of such approach. SURPASS [25,51] is based on the fundamental idea that three-dimensional structures of globular proteins (and also the majority of membrane proteins) are to a large extent defined by the local ordering of polypeptide chains, defined as the secondary structure. A united residue of the SURPASS chain represents a single amino acid unit, located in the center of mass of four consecutive alpha carbons. The choice of the four-residue averaging of united residue positions is not incidental. For regular elements of secondary structure, such as helices or β-strands (but also, to a lesser extent, other secondary structure fragments), such averaging leads to a very simple, almost linear, representation of polypeptide chains (Figure 5). On this level of protein-drawing, it is possible to derive several simple statistical rules controlling three-dimensional structures [25,51]. For instance, helical fragments can be treated as linear chains composed of partially overlapping thick spheres, while β-strand fragments are thinner, with the asymmetric shape of their excluded volume profiles, representing differences in distances between neighboring strands belonging to the same β-sheets, and between strands of different sheets. Characteristic distances between helices and β-sheets, and other structural regularities, can also be easily incorporated in the statistical force field of the model. In its simplest version, the SURPASS force field is based on secondary-structure assignment (or predictions), as the only sequence-defined variable-defining interaction patterns. The solvent effects, similarly as to what was assumed in the CABS model, are treated in an implicit fashion. Reconstruction of more detailed structure representations from SURPASS models is not trivial, but, as tested on a large set of globular structures, it is possible, and does not create drastic inaccuracy of the resulting atomistic models. In this context, the structural accuracy (resolution) of the model is in the range of 2–3 angstroms. This is an acceptable range for all-atom structure refinement protocols [52]. The dynamics of SURPASS proteins is simulated by Monte Carlo dynamics schemes within single = trace or replica-exchange (Replica Exchange Monte Carlo (REMC) dynamics) runs. The simplicity of the model allows simulations of much larger systems than accessible for higher-resolution models [25,51]. As shown for representative sets of single-domain globular proteins, even for relatively large proteins, a single REMC run provides trajectories covering the entire conformational space of the model, with multiple visits of native-like structure regions. Due to its computational speed and the fact that its low-resolution results can be used as the starting frames for higher-resolution simulations, SURPASS modeling opens up a possibility for efficient multiscale modeling of the long-time dynamics of large proteins and protein systems.

### 2.4. Elastic Network Models

In the past decades, a large number of studies have shown the usefulness of NMA and Principal Component Analysis (PCA) for the prediction and analysis of protein co-operative motions [13,14,53,54,55,56,57,58,59,60,61,62]. Many of these studies were conducted using computationally simple ENM that use uniform harmonic potentials for interacting atom (or residue) pairs instead of more complicated potentials. 

In the early 1980s, the NMA method was adapted by Go [63], Karplus [64], and others for studies of thermal fluctuations of proteins around their native structures. PCA and NMA of proteins and other biomolecules applied to their equilibrium dynamics around native states have shown that low-frequency normal modes correspond to the co-operative motions of large parts of the protein structure that are essential for protein function, while high-frequency modes corresponding to the isolated, non-co-operative vibrations of small parts of the structure are functionally meaningless. The long-range character of low-frequency modes enables allosteric interaction of various parts of proteins that are spatially distant from one another. The old paradigm that the protein sequence encodes protein structure, and the structure determines protein function has been modified in the last 20 years by a new paradigm assuming that sequence encodes the structure, the structure determines protein dynamics, and the dynamics encodes protein function. It has also been shown that, for globular proteins, their dynamics is strongly determined by protein shape [65]. Besides describing the near-native motion of globular proteins, PCA can also be applied to structure snapshots from molecular dynamics or Monte Carlo simulations, to structures from nuclear magnetic resonance ensembles (NMR) [66], or to the analysis of large sets (clusters) of protein or RNA structures determined by X-ray crystallography, such as, for example, a set of over 350 HIV-1 protease structures [67].

ENMs have become extremely successful in biology after their reformulation as the Gaussian Network Model (GNM) of proteins [54,68]. GNM assumes that each residue of a protein is represented by a single node, with its co-ordinates usually given by co-ordinates of the Cα atom, although different definitions of nodes (e.g., the center of mass of a side chain) are also frequently used. Nodes that are separated by less than a cut-off distance (usually assumed to be around 7 Å) are connected by identical harmonic springs. Nodes that are further than a cut-off distance are not connected by springs. The model is based on the assumption of Tirion [69,70] that bonded and nonbonded contact interactions in biological structures can be described by a single uniform spring constant parameter. Although such approximation seems, at first, rather unrealistic, the results of normal-mode analysis are almost undistinguishable from the results derived using sophisticated energy-function potentials, such as the CHARMM force field (Chemistry at HARvard Macromolecular Mechanics) [71]. The excellent performance of springlike potentials for normal-mode analysis is due to the fact that any complicated potential function is very well approximated around its minimum by a harmonic function. The basic difference between elastomeric polymer networks and proteins is that a protein is a densely packed collapsed heteropolymer with a unique packing of amino acids and many nonbonded interactions, whereas for polymers these nonbonded interactions are not important. The fluctuations of residues in proteins computed with GNM agree surprisingly well with experimental temperature factors in the Protein Data Bank (PDB). Excellent agreement is typically achieved by this approach when crystallographic B-factors [68] or H/D exchange data [72] are plotted against mean-square fluctuations computed with GNM as a function of the residue index. Additionally, the covariance of fluctuations of two residues *i* and *j* that provides information about correlations of equilibrium fluctuations of residues in protein structure can be simply computed with elastic network models. Pairs of residues that have large values of such correlations may communicate with each other because of functional reasons, for example, by participating in common allosteric communication paths. In the GNM, it is assumed that fluctuations of residues around their mean positions are spherically symmetric. 

The Anisotropic Network Model (ANM) assumes that fluctuations of residue positions around their starting points are, in contrast to the spherical approximation of GNM, anisotropic and are represented by ellipsoids. The model can be used to calculate normal modes from a single structure [73,74]. The cut-off distance for defining the springs is usually assumed to be 13 Å that is significantly larger than the cut-off distance of around 7 Å that is usually used in GNM. For most of the proteins in PDB, crystallographic B-factors are scalars and GNM very well predicts the magnitude of fluctuations of residues. Protein structures solved with high or ultrahigh resolution (less than 0.8 Å) usually have anisotropic B-factor tensors deposited in PDB, and ANM that predicts the tensor of anisotropic B-factors is a very useful tool for modeling proteins with very high resolution reference structures (Figure 6).

Most co-operative (lowest-frequency) modes of motion are almost insensitive to the details of the model and parameters, and can be satisfactorily captured by adopting low-resolution ENMs. The idea was applied by Doruker and Jernigan [75] to low-resolution EN representation, where each node stands for a group of m residues condensed into a single interaction site. Calculations performed for different levels of coarse-graining (*m* = 1, 2, 10, 40) have shown that the slowest modes of motion are preserved regardless of the level of coarse-graining. Mixed coarse-graining has also been introduced for EN models, where a protein’s native conformation is represented by different regions having high and low resolution [76]. The aim here was to capture the dynamics of the interesting parts in structures at higher resolution and retain the remainder of the structure at lower resolution, keeping the total number of modes sufficiently low for computational tractability.

Except as mentioned above, ENMs have been successfully used in many combinations with different input information and methods, such as, for example, data from electron microscopy [14], atomistic MD simulations [17,77], Brownian simulations [78], structure-based models [79,80], and many other combinations that have been thoroughly reviewed [13,14,53,54,55,56,57,58,59,60,61,62].

CABS and SURPASS obviously have several limitations, characteristic for all CG models of low and moderate resolution. First, in multiscale simulations the shift from atom level to CG representations is easy (unique mapping), while the reverse projection from CG to the atomistic representation is always nonunique and less accurate, and needs to be carefully performed [8]. Knowledge-based force fields, while working very well for “typical systems”, are also not easy-to-use in studies of rare and more complicated events as, for instance, structure prediction for proteins adopting different secondary structures for the same or very similar sequence fragments. It should be, however, noted that the secondary-structure biases in properly designed knowledge-based statistical potentials (as in CABS or SURPASS models) are weak, and may compete with other forces, allowing quite realistic simulations of long-time dynamics and conformational transitions [8]. In this context, the use of knowledge-based force fields in CG simulations is less restrictive than use of reference structures in elastic network models.

## 3. Applications of Coarse-Grained Modeling: ENM and CG Monte Carlo Simulations

### 3.1. Modeling of the Structural Flexibility of Folded Proteins

When the range of protein structural dynamics studies can be limited to a roughly defined vicinity of a specific folded structure (determined by experiment), ENM and CG models or their modifications can be very effective in predicting protein flexibility [8,36,81,82,83].

In the past decades, ENM-based modeling has definitely been the most common approach to modeling fluctuations near the input folded structures. There are many available methods based on ENM, some of them as easy-to-use web servers [67,84,85,86,87,88,89,90,91,92,93,94,95]. CG methods using more sophisticated potentials that are not derived from the input structure are much less common in predictions of protein fluctuations [8,36,37,39,96,97]. These include the CABS-flex method using the CABS CG model for MC simulations of protein flexibility [37,39] near a predefined reference structure that can be taken from experiments or predicted theoretically. The CABS-flex method has been shown to provide a consistent view of protein flexibility with short timescale MD simulations [36] and NMR ensembles [38]. While qualitatively similar to ENM predictions, CABS-flex generated better correlated pictures of protein fluctuations (on average for the studied set) in comparison to all-atom MD simulations [36]. The CABS-flex method is currently available as a web server (an improved 2.0 version was recently published [39]) and a standalone package [40]. 

Below, in Figure 7, we present the comparison of fluctuation profiles (root mean square fluctuation (RMSF) for protein 1hpw (using the first model from the 1hpw PDB ensemble as the input) obtained from:
NMR ensemble: data calculated using 10 models deposited in the PDB code: 1hpw.MD all-atom simulation: data obtained using a 10 nanosecond trajectory with an AMBER8.0 force field taken from the MoDEL database of MD trajectories [38]; RMSF was calculated for the entire trajectory consisting of 10,000 models.CG simulation using the CABS model: data obtained using the CABS-flex 2.0 web server [39]; results calculated using the default server settings; RMSF was calculated for the set of 10 representative models (obtained by a cluster analysis of 10,000 snapshots) from the simulation trajectory.CG simulation using the SURPASS model: data obtained with the following SURPASS [25] settings: isothermal MC simulation in low reduced temperature (*T* = 0.2), 10,000 MC steps, 1 and 3-bead motions; RMSF was calculated for the entire trajectory of 100 models.ENM modeling: data obtained using the DynOmics web server [85,95] that integrates two ENM methods: the GNM and the ANM, calculated using default server settings; real time calculation: <1 min; RMSF was calculated for the set of 20 models based on the 10 slowest modes (2 models per mode for extreme positions during movement); the amplitude of motion along each mode was chosen so that the RMSD (root-mean-square deviation of atomic positions) between the models was less than 2Å; all models were generated using the ‘Molecular Motions—Animation’ option available on the results page of the DynOmics server.

Figure 7 shows that ENM and CABS MC simulations provide dynamic profiles closest to experimental characteristics (such experiments have been made for many protein systems, see reference [38] for more details). The deeply CG SURPASS method is less accurate, probably mostly due to its extremely simplified interaction model, although dynamic profiles are quite realistic. Obviously, CG methods are computationally much less expensive than atomistic simulations by MD. Results presented in Figure 7, while quite illustrative, should be treated with some caution. For instance, an NMR ensemble certainly does not show the full picture of protein flexibility, and MD simulations are probably too short to sample all possible conformations.

The ENM-based exploration of protein conformational space near the given folded structure can also be practically used in the refinement of models derived from protein-structure prediction methods [98,99,100,101,102,103,104]. For example, Feig proposed and successfully tested a modeling scheme based on NMA and MD simulations with constraints and an efficient sampling scheme [105,106]. The procedure used by Feig’s group was an iterative one. First, they performed molecular mechanics energy minimization, and then employed NMA computations around the local energy minimum. Subsequently, they generated and evaluated an ensemble of possible new conformations along the lowest-frequency normal modes. Normal modes provide guidance toward the effective refining of protein structures. A similar approach was used by Tama and Brooks to refine cryo-Electron Microscopy structures [107]. Similarly to ENM and related models, CG Monte Carlo simulations can also be easily combined with MD simulations [108,109,110,111].

### 3.2. Modeling Large-Scale Structural Transitions

ENMs have been shown to be useful to study large-scale protein conformational transitions in cases when starting and ending conformational states of the transition are available [14,53,96,112,113,114,115,116,117,118]. Protein conformational transitions frequently occur in biology, and usually relate to protein function. The most frequently occurring large-scale conformational transition involves two protein conformations: the “open” and the “closed” form, which are the starting and the ending states of the transition. Assuming that the physical mechanism of normal modes is used in nature to drive large-scale conformational transition, an ENM can be used to model such transitions using both the “open” and the “closed” structures [112]. Since such large-scale conformational transitions are frequently slow and difficult for detailed modeling with MD simulations, the above-mentioned ENM approach represents a very useful and computationally inexpensive alternative method to study them. Both approaches usually give similar results [77]. ENM-based methods have been successfully applied to structural transitions of very large molecular structures [14], such as a ribosome [119,120,121]. 

CG protein simulation models, such as CABS or SURPASS, offer an alternative approach to predicting large-scale transitions that do not require starting and ending conformational states. The CABS model has been used for simulations of folding mechanisms from the fully denatured to the folded state [28,29,30,31,32,43]. This way, with some successes, the CABS model can be used for de novo structure prediction. Obviously, with the growing structural database, de novo modeling of protein structures becomes a rather theoretical challenge with decreasing practical importance. For the majority of new proteins, it is now possible to identify close or distant homologs with known, experimentally determined structures. CG models, including CABS, have become powerful tools for comparative modeling, especially for the cases of distant homology, where differences between template and target structures are significant. The CABS-fold server can be easily used for de novo structure prediction (purely de novo, without any knowledge about the plausible starting structure, is reasonable only for rather small proteins) and for template/template-based comparative modeling. In a different context, using weak or limited structural restraints, and/or known starting structures, models such as CABS, UNRES, or related models [44,122,123,124,125,126] can be powerful tools for simulations of large-scale structural rearrangements. Folding/unfolding pathways, chaperone effects, large-scale structural rearrangements during molecular docking, and other related processes can be efficiently simulated. The medium-resolution CG models, while opening the timescale about three to four orders of magnitude wider than that accessible for classical MD simulations, still remain computationally too expensive for modeling multiple large-scale structural transitions of large proteins. The SURPASS model opens up such possibility. Example results of SURPASS simulations of folding–unfolding processes in a globular protein are illustrated in Figure 8. The entire conformational space of large proteins can be effectively sampled. This allows very detailed studies of the very long-time dynamics of proteins and protein complexes, although biased by the necessary simplifications of structure representations and the model (knowledge-based, statistical) force field. Although maintaining fundamental features of protein structure, such crude models can be combined with more accurate modeling tools, for example, providing representative starting conformations for de novo simulations with higher resolution tools, or a plethora of random, partially folded and more compact folded/misfolded structures for derivation of realistic statistical potentials for other CG models.

## 4. Concluding Remarks and Perspectives

Computational exploration of protein conformational transitions has already been quite successful in many tasks of structural biology, including rational drug design [4,5,127,128], but it is often very challenging. The major difficulties arise from inexact sets of parameters (e.g., force-field parameters [4,129]), and inability to access long simulation timescales of many important biological processes. In this review, we present two approaches to extend accessible timescales and sizes of modeled systems: ENMs and CG models using MC dynamics. These can be combined with atomistic-level MD, providing efficient strategies of multiscale simulations of proteins and protein complexes. While the ENM approach is well suited for speeding up collective motions, CG MC can be applied to a broader range of functional moves. 

ENM- (or NMA)-based methods are well suited only for certain problems [54]. They can describe well protein dynamics around the known experimentally solved native state, or conformational transitions between two states, such as “open”–“closed”, or “bound”–“unbound”, for which both structures are known. The applicability of ENM methods strongly depends on how collective the motion is [53,130]. In other words, ENM-based methods do not provide real dynamics and real transition pathways, but only an approximation based on the combination and interpolation of normal modes. The high accuracy of such a simplified approach for the description of system dynamics near the given reference structure clearly shows that the local dynamics is, to a large extent, controlled by the general structural features of proteins, and is less dependent on specific interaction patterns between amino acid residues. For example, the use of ENM in protein-ligand docking is restricted to selected cases in a few only collective motions responsible for binding a ligand [130]. In this context, CG MC simulations appear to be more realistic. In principle, simulation methods, such as MD and CG MC (or CG MD), can be used for simulations of entire folding/unfolding processes, although the cost (and limitations of practical applicability) of such simulations rapidly grows with increasing resolution of the models. Deeply CG models, such as SURPASS, discussed here, allow simulations of the largest systems, although such CG levels should be used in connection with more accurate modeling tools.

In the above context, the future will see combinations of ENM and other CG methods of various resolution with atomistic MD [14,131,132,133]. Such hierarchical integrative approaches should be properly tailored for specific problems and utilize structural data from available experimental sources, such as X-ray crystallography, NMR, EM, and SAXS [134,135,136]. The challenges of integrative modeling call for computational tools that are available as easy-to-use software packages based on different kinds of data and merging with other tools, just as the CABS model described here [40].

## Figures and Tables

**Figure 1 ijms-19-03496-f001:**
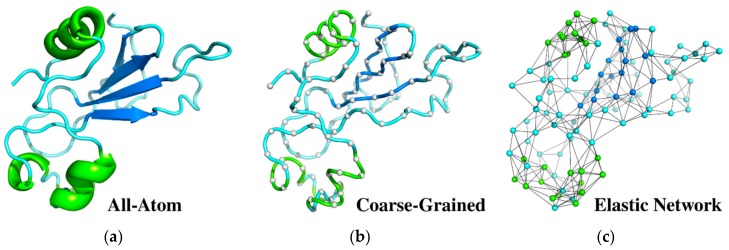
Different representations of protein structure (identifier of entry in the Protein Data Bank (PDB code): 1a2p) used as a reference state for structure dynamics studies. (**a**) All-atom structure used in molecular dynamics (MD) simulations shown as a ribbon diagram; (**b**) deeply coarse-grained model (gray balls represent a single center of interaction per residue connected by a tube) for Monte Carlo (MC) dynamics simulations; (**c**) coarse-grained elastic network model with spring-type constraints and nodes in the positions of Cα atoms. Colors refer to the secondary structure assignment of protein fragments; helices are shown in green and β-strands are in dark blue.

**Figure 2 ijms-19-03496-f002:**
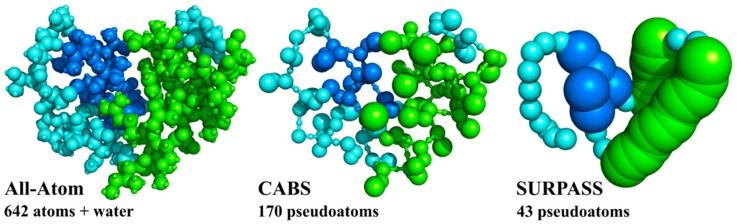
All-atom (AA) and two coarse-grained (CG) representations of protein structures. For a small protein (PDB code: 1CRN), consisting of 46 residues, AA representation needs the explicit treatment of 642 atoms (and a large number of solvent atoms when an explicit solvent model is analyzed). A lower-resolution CG CABS model (CA, CB, Side chain model) requires explicit treatment of 170 pseudoatoms and the deeply CG Single United Residue per Preaveraged Secondary Structure fragment (SURPASS) model reduces the number of simulated pseudoatoms (or rather pseudoresidues in this case) to just 43, reducing simulation cost by orders of magnitude (Figure 3). In both cases of CG representation, the atomistic structure can be approximately reconstructed thanks to the properly defined geometry and interaction schemes of these models.

**Figure 3 ijms-19-03496-f003:**
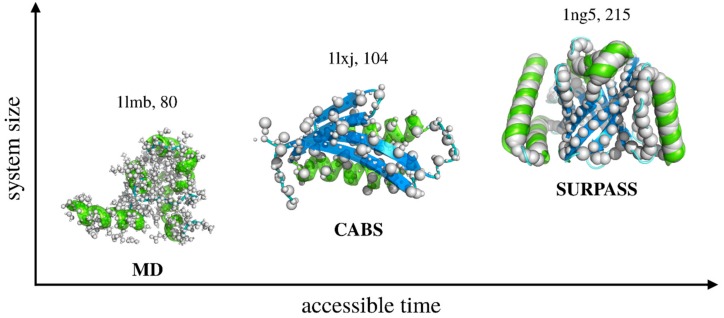
Increasing range of applicability of dynamics modeling, from MD to deeply coarse-grained simulations. CABS (assuming C-Alpha, C-Beta, and side-chain representation of polypeptide chains) is a moderate resolution model of protein structure and dynamics. SURPASS is a deeply CG model, where single united residues represent entire amino acids. Both CG structure models use Monte Carlo dynamics sampling for fast simulations of long-time protein dynamics.

**Figure 4 ijms-19-03496-f004:**
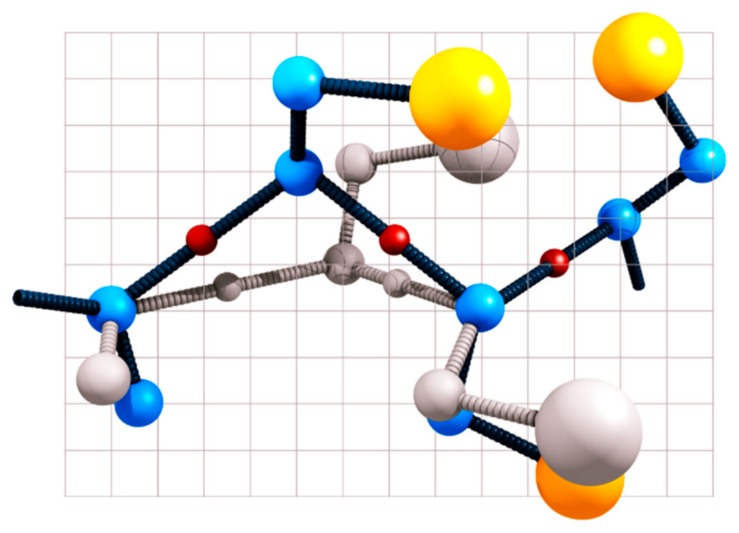
CABS representation of a small protein chain fragment. Positions of Cα pseudoatoms are restricted to grid nodes of the underlying simple cubic lattice with spacing of 0.61 Å (for more details see Reference [8,24]). Allowing small fluctuation of the expected Cα–Cα distance enables hundreds of possible orientations of these pseudobonds. Side-chain pseudoatoms (Cβ in blue and an additional center of the side chain, where applicable, in yellow) are not restricted to the lattice and their positions are defined by the geometry of the main chain and statistical properties of specific amino acids. A move of a single Cα leads to specific displacements of three side chains (new positions of the side chains are shown in gray). Thanks to the high-coordination lattice representation of the main chain geometry, all possible local moves are stored in large precalculated data tables.

**Figure 5 ijms-19-03496-f005:**
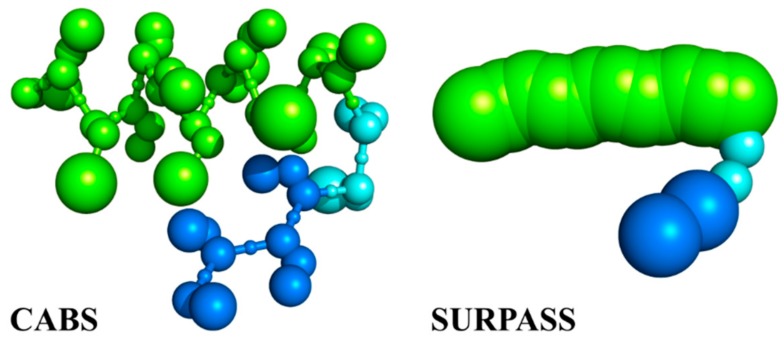
Comparison of CABS and SURPASS CG representations of proteins. A short structural element containing helical (green) and β-strand (blue) fragments connected by a loop.

**Figure 6 ijms-19-03496-f006:**
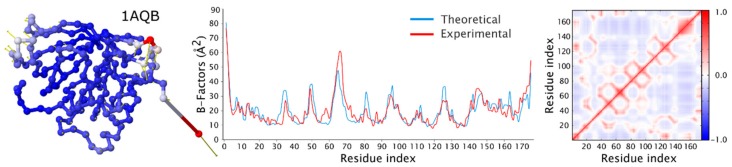
Evaluation of fluctuational dynamics of retinol binding protein (PDB code: 1aqb, chain A) using DynOmics server. First panel shows protein structure with residues represented by balls located at the positions of Cα atoms and colored according to the values of B-factors (with red representing highly mobile and blue low mobile residues). Yellow arrows show the direction of oscillation. The chart in the middle compares experimental B-factors (in red) with predicted theoretical values (in blue) as a function of residue index. The effective force constant of the GNM springs is 0.92 k_B_TÅ^−2^, and corresponding rescaling prefactor is 85.40. Chart on the right shows cross-correlations (CC) between residue fluctuations with the coloring scheme shown on the right (with red denoting highly correlated and blue highly anticorrelated motions).

**Figure 7 ijms-19-03496-f007:**
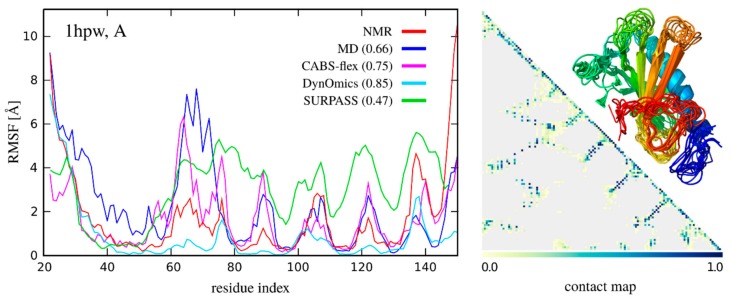
Comparison of residue fluctuation (root mean square fluctuation (RMSF)) profiles for an example protein (PDB code: 1hpw, chain A). Chart on the left presents RMSF values (in Å) derived from NMR ensembles (red) and simulation trajectories: all-atom MD (blue), CABS-flex (pink), DynOmics (light blue), and SURPASS (green). In the legend, the values in brackets show Spearman’s correlation coefficients for residue fluctuations between NMR and each method. The chart on the right presents a contact map (frequency of contacts is showed in different colors, see the color scale) and example fluctuation profiles visualized on the structures given by the CABS-flex 2.0 server [39].

**Figure 8 ijms-19-03496-f008:**
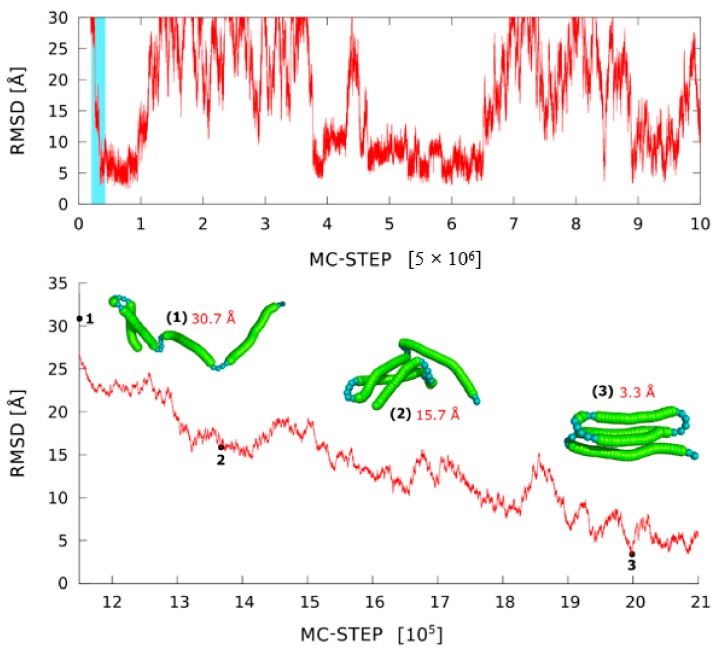
MC dynamics trace for a single replica for alpha protein (PDB code: 1k40) in SURPASS representation. The lowest panel shows the enlargement of the coil-to-globule transition taken from a small fragment of the RMSD trajectory marked in blue in the upper plot. The selected snapshots of protein structures illustrate the observed mechanism of fold assembly.

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
