# Peer review of "Modeling of Protein Structural Flexibility and Large-Scale Dynamics: Coarse-Grained Simulations and Elastic Network Models"

_ijms, 2018, doi:10.3390/ijms19113496_

Reviewer 1 Report

In this review, the authors describe coarse-grain modeling to characterize protein structure and dynamics. Specifically, they focus on CABS and SURPASS representations coupled with Monte Carlo-based sampling, and on elastic network models. The text is overall well written and focusses on an interesting topic, that I believe will be of interest for the readers of IJMS. The points below should however be addressed.
1) The authors refer several times in the text to “classical molecular dynamics”, which I find slightly confusing. I assume the authors mean molecular dynamics using an all-atom representation. I suggest stating that explicitly in the text.
2) The text is at times vague. I refer to sentences mentioning system sizes and timescales (e.g. lines 40-42, 74-75, or 217-218). Authors should be more explicit in what they consider “long timescales” and “large systems”. As side note, it should be considered that all-atom MD simulations being multi-microsecond long, and simulations of systems with hundreds of thousands of atoms, are now commonplace. While often insufficient to observe spontaneous folding from an unfolded state, or ligand binding, these timescales and sizes do allow addressing “interesting protein systems (especially from the medical point of view)” (lines 42-43).
3) Descriptions of limitations of described methods appears to be almost entirely lacking. Especially since this is a review, I find if would be particularly useful to mention them explicitly. For instance:
a. If I understand correctly, SURPASS does not allow simulating changes in secondary structure.
b. Knowledge-based force fields are usually biased towards folded structures (if knowledge is obtained from databases of protein X-ray structures). Unfolded/disordered regions are usually not as well represented.
c. Elastic network models are, as hinted in the text by authors, limited to the characterization of a specific stable conformation or, in some cases, can characterize the transition between multiple known states.
d. Monte Carlo sampling is excellent at generating statistical ensembles of a system under study but is less appropriate to yield information about dynamics within a system (e.g. studying allosteric effects).
4) I find results shown if Figure 7 not totally convincing. These are my main concerns:
a. Experimental data used as comparison is an NMR ensemble. Such an ensemble is not a perfect representative of the protein’s conformational space, as it is usually just a collection of models that are individually consistent with the data (i.e., the ensemble is Boltzmann-averaged).
b. The MD simulation used as comparison is 10 ns long. This is likely too short for the protein to have fully sampled its conformational space, which might affect matching against experimental data. I should point out that recent all atom force fields such as Amber ff14SB or CHARMM3 (what was used in the described simulation?) do a remarkable job in reproducing NMR observables such as s2, so seeing a CG force field performing better than an atomistic one is a bit surprising.
c. A selection of “representative structures” is mentioned. The authors should indicate how such representatives were selected, as different selections will yield different RMSF.
5) Authors mention that the models discussed in this review are suitable for multiscale approaches. Multiscale modelling requires backmapping (i.e. switching from a CG to an atomistic representation), which is a problem becoming increasingly hard, the higher the coarse-graining level. If the authors want to make the point that these methods are useful for multiscale modelling, I suggest mentioning backmapping with either references to works having made efforts in this direction or describing it explicitly.
6) Mentioning execution time only makes sense if information about used computer architecture is provided (lines 329-335).
7) Caption in Figure 4 mentions a mesh grid of 0.61 Angstrom. This seems an awfully precise number, is this the result of some profiling? If so, a reference to the article deriving this number should be added.
8) Lines 218-220: any reference?

Author Response

In this review, the authors describe coarse-grain modeling to characterize protein structure and dynamics. Specifically, they focus on CABS and SURPASS representations coupled with Monte Carlo-based sampling, and on elastic network models. The text is overall well written and focusses on an interesting topic, that I believe will be of interest for the readers of IJMS. The points below should however be addressed.

Response: We appreciate the high opinion of the Reviewer.

Point 1:The authors refer several times in the text to “classical molecular dynamics”, which I find slightly confusing. I assume the authors mean molecular dynamics using an all-atom representation. I suggest stating that explicitly in the text.

Response 1: It has been clarified.

“In contrast to classical, all-atom, Molecular Dynamics the modeling strategies discussed here allow quite accurate modeling of much larger systems and longer time dynamic phenomena.”

 Point 2: The text is at times vague. I refer to sentences mentioning system sizes and timescales (e.g. lines 40-42, 74-75, or 217-218). Authors should be more explicit in what they consider “long timescales” and “large systems”. As side note, it should be considered that all-atom MD simulations being multi-microsecond long, and simulations of systems with hundreds of thousands of atoms, are now commonplace. While often insufficient to observe spontaneous folding from an unfolded state, or ligand binding, these timescales and sizes do allow addressing “interesting protein systems (especially from the medical point of view)” (lines 42-43).

Response 2: The text has been modified as suggested.

In practice, many of the most interesting protein systems (especially from the medical point of view) are either too large to be effectively simulated using MD or require very large supercomputer resources.” (lines 42-43)

“Important biomacromolecular systems can be very large (consisting of many atoms) and the timescale of interesting processes can be beyond the MD simulation range accessible to contemporary computers.” (lines 74-75)

“Simplicity of the model allows very fast simulations of much larger systems than accessible for higher resolution models [25,52].” (lines 219-220)

 Point 3: Descriptions of limitations of described methods appears to be almost entirely lacking. Especially since this is a review, I find if would be particularly useful to mention them explicitly. For instance: a. If I understand correctly, SURPASS does not allow simulating changes in secondary structure. b. Knowledge-based force fields are usually biased towards folded structures (if knowledge is obtained from databases of protein X-ray structures). Unfolded/disordered regions are usually not as well represented. c. Elastic network models are, as hinted in the text by authors, limited to the characterization of a specific stable conformation or, in some cases, can characterize the transition between multiple known states. d. Monte Carlo sampling is excellent at generating statistical ensembles of a system under study but is less appropriate to yield information about dynamics within a system (e.g. studying allosteric effects).

Response 3: We added a paragraph in which the major limitations of the discussed methods are outlined. 

CABS and SURPASS have obviously several limitations, characteristic for all CG-models of low and moderate resolution. First, in multiscale simulations the shift from atom level to CG representations is easy (unique mapping) while the reverse projection from CG to the atomistic representation is always non-unique and less accurate, and needs to be carefully performed [8]. Also, the knowledge-based force fields, while working very well for “typical systems” are not easy-to-use in studies of rare and more complicated events, as for instance structure prediction for proteins adopting different secondary structures for the same or very similar sequence fragments. It should be however noted that the secondary structure biases in properly designed knowledge-based statistical potentials (as in CABS or SURPASS models) are weak and may compete with other forces, allowing quite realistic simulations of long time dynamics and conformational transitions [8]. In this context the use of knowledge-based force fields in CG simulations is less restrictive than use of reference structures in elastic network models.” (lines 308-319)

 Point 4: I find results shown if Figure 7 not totally convincing. These are my main concerns: a. Experimental data used as comparison is an NMR ensemble. Such an ensemble is not a perfect representative of the protein’s conformational space, as it is usually just a collection of models that are individually consistent with the data (i.e., the ensemble is Boltzmann-averaged). b. The MD simulation used as comparison is 10 ns long. This is likely too short for the protein to have fully sampled its conformational space, which might affect matching against experimental data. I should point out that recent all atom force fields such as Amber ff14SB or CHARMM3 (what was used in the described simulation?) do a remarkable job in reproducing NMR observables such as s2, so seeing a CG force field performing better than an atomistic one is a bit surprising. c. A selection of “representative structures” is mentioned. The authors should indicate how such representatives were selected, as different selections will yield different RMSF.

Response 4: We clarified the fragment of the text, referring to the data presented in Figure 7.

“Below, in Figure 7, we present the comparison of fluctuation profiles (RMSF, root mean square fluctuation) for protein 1hpw (using the first model from the 1hpw PDB ensemble as the input) obtained from:

·            NMR ensemble: data calculated using 10 models deposited in the PDB code: 1hpw

·            MD all-atom simulation: data obtained using a 10 nanosecond trajectory with AMBER8.0  force field taken from the MoDEL database of MD trajectories [38]; RMSF was calculated for the entire trajectory consisting of 10,000 models

·            CG simulation using the CABS model: data obtained using the CABS-flex 2.0 web server [39]; results calculated using the default server settings; real time calculation: 22 minutes; RMSF was calculated for the set of 10 representative models (obtained by a cluster analysis of 10000 snapshots) from the simulation trajectory

·            CG simulation using the SURPASS model: data obtained with the following SURPASS [25] settings: isothermal MC simulation in low reduced temperature (T=0.2), 10000 MC steps, 1 and 3-bead motions; real time calculation: 15 seconds; RMSF was calculated for the entire trajectory of 100 models

·            ENM modeling: data obtained using the DynOmics web server [86,96] that integrates two ENM methods: the Gaussian Network Model (GNM) and the Anisotropic Network Model (ANM); calculated using the default server settings; real time calculation:<1 minute; RMSF was calculated for the set of 20 models based on the 10 slowest modes (2 models per mode for extreme positions during movement); the amplitude of motion along each mode was chosen so that the RMSD between the models was less than 2Å; all models were generated using the ‘Molecular Motions - Animation’ option available on the results page of the DynOmics server.” (lines 339-360)

“Figure 7 shows that ENM and CABS MC simulations provide dynamic profiles closest to experimental characteristics (such experiments have been made for many protein systems, see reference [38] for more details). The deeply CG SURPASS method is less accurate, probably mostly due to its extremely simplified interaction model, although the dynamic profiles are quite realistic. Obviously the CG methods are computationally much less expensive than atomistic simulations by MD. Results presented in Figure 7, while quite illustrative should be treated with some caution. For instance, an NMR ensemble certainly does not show the full picture of protein flexibility and the MD simulations are probably too short to sample all possible conformations.” (lines 370-377)

Point 5: Authors mention that the models discussed in this review are suitable for multiscale approaches. Multiscale modelling requires backmapping (i.e. switching from a CG to an atomistic representation), which is a problem becoming increasingly hard, the higher the coarse-graining level. If the authors want to make the point that these methods are useful for multiscale modelling, I suggest mentioning backmapping with either references to works having made efforts in this direction or describing it explicitly.

Response 5: An appropriate comment has been added to the manuscript.

“First, in multiscale simulations the shift from atom level to CG representations is easy (unique mapping) while the reverse projection from CG to the atomistic representation is always non-unique and less accurate, and needs to be carefully performed [8].” (lines 309-311)

Point 6: Mentioning execution time only makes sense if information about used computer architecture is provided (lines 329-335).

Response 6: The data regarding the execution time has been deleted. Reading of the entire text provides quite accurate information regarding the computational cost of various modeling schemes.

·            “MD all-atom simulation: data obtained using a 10 nanosecond trajectory with AMBER8.0  force field taken from the MoDEL database of MD trajectories [38]; RMSF was calculated for the entire trajectory consisting of 10,000 models

·            CG simulation using the CABS model: data obtained using the CABS-flex 2.0 web server [39]; results calculated using the default server settings; real time calculation: 22 minutes; RMSF was calculated for the set of 10 representative models (obtained by a cluster analysis of 10000 snapshots) from the simulation trajectory

·            CG simulation using the SURPASS model: data obtained with the following SURPASS [25] settings: isothermal MC simulation in low reduced temperature (T=0.2), 10000 MC steps, 1 and 3-bead motions; real time calculation: 15 seconds; RMSF was calculated for the entire trajectory of 100 models” (lines 343-353)

Point 7: Caption in Figure 4 mentions a mesh grid of 0.61 Angstrom. This seems an awfully precise number, is this the result of some profiling? If so, a reference to the article deriving this number should be added.

Response 7: This is just the exact number, assumed during designing of CABS model. The detailed discussion of this choice can be found in the cited references (added).

Figure 4. CABS representation of a small protein chain fragment. Positions of Cα pseudoatoms are restricted to grid nodes of the underlying simple cubic lattice, with spacing of 0.61 Å (for more details  see ref [8,24]).” (lines 133-135)

Point 8: Lines 218-220: any reference?

Response 8: The required references have been added.

“Simplicity of the model allows very fast simulations of much larger systems than accessible for higher resolution models [25,52].” (lines 219-220)

Reviewer 2 Report

In this paper the authors have reviewed the use of Coarse-grained Simulations and Elastic Network Models for larger system and longer time scale. They also compare the application of CG (CABS and SURPASS) and the ENM methods to study the fluctuation and large-scale structural transitions of proteins and discuss their potential advantages and limitations. Overall, this is an important paper and would be interesting to both the experts and the beginners working on molecular dynamics simulations of biomolecules.

Author Response

We appreciate the high opinion of the Reviewer.

Reviewer 3 Report

The manuscript by Kmiecik et al. is a mini-review describing two alternative approaches to study protein molecular dynamics useful to predict conformational transitions in polypeptide chain. The Authors propose an extension of the accessible timescales and size of the modeled systems by combining Elastic Network Models and Coarse Grained simulations in MC dynamics. This approach should provide an additional efficient strategy for study conformational fluctuations in proteins and protein complexes.

Minor points

The use of undefined abbreviations or common saying should be avoided in order to render the manuscript easily readable (see those present in the lines 98 and 101.

Some grammatical errors (as those in lines 112, 153, 312) as well as punctuation usage (as those in lines 134 and 265) should be corrected.

Author Response

We appreciate the high opinion of the Reviewer.

Point 1: The use of undefined abbreviations or common saying should be avoided in order to render the manuscript easily readable (see those present in the lines 98 and 101).

Response 1: It has been corrected.

“Both approaches have their advantages and disadvantages pros and cons, but since proteins (and generally biomacromolecules) can be characterized by many specific features and owing to the rapidly growing databases of structural biology, statistical force fields can be a good choice. Consequently, instead of using various ersatz MD molecular dynamics schemes we prefer properly designed Monte Carlo dynamics sampling.” (lines 98-102)

Point 2: Some grammatical errors (as those in lines 112, 153, 312) as well as punctuation usage (as those in lines 134 and 265) should be corrected.

Response 2: We addressed all of indicated errors and carefully checked the entire text to make language corrections suggested by all reviewers, especially in lines proposed by Reviewer 3.

“How costly it is to simplify the representation depends not only on the assumed level of coarse-graining, but also on the force field model and sampling schemes used in simulations.” (lines 112-114)

“On the contrary, contacts of charged residues in the central regions of a globule, where parallel and antiparallel orientations of side chains in contact are possible, are extremely rare, and therefore the antiparallel contacts of charged residues are treated as repulsive (unlike).” (lines 153-155 and further presence in the text)

“There are many available methods based on ENM, some of them as easy-to-use web servers [68,85–96].” (line 312, now line 328 and further presence in the text)

“Positions of Cα pseudoatoms are restricted to grid nodes of the underlying simple cubic lattice with, spacing of 0.61 Å (for more details see ref [8,24]).” (line 134)